# Effects of Biochar Application and Irrigation Methods on Soil Temperature in Farmland

**Yanhong Ding [1,2], Xiaoyu Gao [1], Zhongyi Qu [1,\*], Yonglin Jia [1], Min Hu [1] and Changjian Li [3]**

[1]  College of Conservancy and Civil Engineering, Inner Mongolia Agricultural University, Hohhot 010018, China; dyhjxfd@126.com (Y.D.); gaoxiaoyu000@163.com (X.G.); jiayonglin1611@163.com (Y.J.); humin1994@126.com (M.H.)
[2]  College of Water Resources and Architectural Engineering, Northwest A&F University, Yanglin 712100, China
[3]  College of Water Resources &Civil Engineering, China Agricultural University, Beijing 100083, China; lichangjian0714@163.com
\*  Correspondence: quzhongyi@imau.edu.cn; Tel.: +86-471-4316865

**Abstract:** Soil temperature plays a vital role in determining crop yield. Excessive irrigation may result in low soil temperature and a waste of water resources. In this paper, field experiments were carried out to evaluate the influence of irrigation methods and biochar application on soil temperature. The experiment included six treatments: (a) YB: biochar application in border irrigation with Yellow River water; (b) GB: biochar application in border irrigation with groundwater; (c) DB: biochar application in drip irrigation with groundwater; (d) Y(CK): border irrigation with Yellow River water; (e) G(CK): border irrigation with groundwater; (f) D(CK): drip irrigation with groundwater. The results are as follows: coupling drip irrigation and biochar, soil temperature increased by 1.20–3.87%. In the biochar application in border irrigation with Yellow River water and groundwater, soil temperature increased by 0.80–2.40% and 1.01–5.15%, respectively. Biochar is a medium for reducing the heat exchange of soil and atmosphere, as it hinders bi-directional heat movement. This mechanism was especially apparent at a 0–10 cm soil depth in the treatments of border irrigation using Yellow River water and groundwater. Biochar may help stabilize the fluctuation of soil temperature and improve the soil accumulated temperature. The effect of drip irrigation at 5–10 cm depth, border irrigation using the groundwater and the Yellow River water was great on soil temperatures above the 10 cm level but less on deep soil temperatures. After applying biochar to soil, the soil temperature was more sensitive to external temperature changes, such as air temperature and water temperature. Therefore, in the Hetao irrigation area, applying a proper amount of biochar to farmland soil was shown to improve the water and heat environment and improve the effectiveness of traditional border irrigation in synchronizing water and heat, especially under the drip irrigation condition. The results here suggest that using biochar under drip irrigation can promote growth and increase yield.

**Keywords:** different irrigation methods; biochar; soil temperature

## 1. Introduction

Soil is an important part of the ecosystem; it directly impacts the plant world and their surrounding climate. Soil temperature plays an important role in the climate system [1]. Plant growth and crop production are known to largely depend on soil temperature [2,3]. Some previous research has proven that soil temperature is an important factor for crop growth, soil respiration, water-salt migration in soil [4–6], the amount and activity of soil microorganisms, greenhouse gas emissions and soil carbon stock. Many of these processes, in turn, impact seed germination and crop yields [7–10]. Soil

temperature affects the biochemical cycles of soil carbon (C), nitrogen (N), and many other elements, thus affecting soil quality, plant growth, and crop yield [11]. Zhou et al. [12] and Shen et al. [13] proved there is a correlation between temperature increase and crop growth, hence, controlling soil temperature is of great importance for obtaining high yields. However, the low atmospheric temperature always results in lower soil temperature in northern China, which may influence germination and growth in the early stage of crops, and farmers implement conventional measures, such as straw mulching, tillage, and other methods. However, soil temperature is a dynamic soil property, as it changes with soil moisture, solar energy, solar radiation, landscape and so on, and those measures are time-consuming and costly. Therefore improving the soil heat flux by changing water distribution and enhancing soil temperature to obtain more crop yield in northern farmland has gradually become a concern of researchers.

Solar radiation is the main source of soil temperature, which is actually a process of transformation and redistribution of net solar radiation energy incident on the land surface. The net radiation exchanges heat with the atmosphere through sensible and latent transfer between the ground and the air, while the other part exchanges heat with the soil through soil heat conduction. The variation of the soil-water environment may cause different soil temperature conditions.

While irrigation is the main factor affecting soil moisture in arid and semiarid regions, soil moisture directly affects the distribution ratio of net radiation energy [14]. This paper discusses ways to change soil heat flux by changing water distribution, and improving soil temperature in order to gain higher crop yield. We also compared the effects of three irrigation methods, which are commonly used to increase soil temperature in China's northern farmlands.

Different irrigation methods lead to different amounts of water entering the soil, which changes the distribution of water in the soil. An appropriate amount of water can increase the specific heat capacity of the soil and reduce the loss of soil temperature. Guo-Hua et al. [15] reported that the surface temperature of sprinkler irrigation and drip irrigation was lower than that of ground border irrigation. The highest soil temperature under drip irrigation was distributed at the 20 cm soil layer, since the high irrigation frequency hindered heat transmission to the deep soil. Sun et al. [16] reported that drip irrigation under mulch had a rapid effect on air temperature change, and soil temperature under drip irrigation was higher at the seedling stage of maize than that of surface irrigation due to the small quantity of irrigation applied. According to Cai-Xia et al. [17], the average soil temperature with alternative furrow irrigation was 0.02~7.00 °C higher than that with conventional furrow irrigation under the condition of the same row direction.

In recent years, biochar has increasingly attracted attention due to its potential contribution to carbon sequestration, emission reduction, and the improvement of crop yield [18,19]. Biochar is produced by the thermal degradation of organic materials in the absence of oxygen, and it can reduce ground reflection, absorb more solar energy, and then rapidly raise soil temperature, due to the black colour. There has been strong evidence of biochar application improving soil moisture and significantly reducing sensitivity of soil respiration to changing temperature [20,21]. Oguntunde et al. [22] reported that on land covered with charcoal in Ghana, surface albedo was reduced by 37% on charcoal-site soils, while soil-surface temperature increased by 4 °C on these sites compared to sites with no charcoal covering, due to the aggravated soil colour of charcoal. Genesio et al. [23] found that surface albedo decreased by up to 80% after the application of biochar to the control sites in bare soil conditions, at a rate of 30 t ha$^{-1}$. While this difference tended to decrease during the crop growing season, after the post-harvesting tillage, the soil treated with biochar again showed a lower surface albedo value (<20–26%) than the control.

Despite these studies on the effect of biochar on soil physical, chemical, and biological properties as well as surface albedo, research on how biochar impacts soil temperature using different irrigation methods is limited. Wang et al. [24] (suggested that biochar increases the soil's moisture content and heat capacity while reducing its temperature range. In Inner Mongolia, large areas of farmland use flooding irrigation, but with excessive irrigation soil temperature reduces sharply. To address

the low soil temperature problem brought on by flood irrigation in the Hetao Irrigation District, soil amendments, such as biochar, offer a promising solution.

In depth research done on the effect of different farming methods, irrigation methods and biochar application on soil temperature suggest there is a significant difference in the soil hydrothermal environment between border irrigation and drip irrigation methods due to variations in the quantity of water used as well as in the irrigation time. There is ample evidence supporting the effect that biochar application has on soil temperature. The effect of coupling irrigation methods and biochar on soil temperature has rarely been reported. Therefore, this study aims to determine the effects of coupling biochar and irrigation methods (the Yellow River water border irrigation, ground water border irrigation and drip irrigation) on soil temperature, which can provide a theoretical basis for the regulation and control of the water and heat environment on farmlands.

## 2. Materials and Methods

### 2.1. Site and Climatic Conditions

The experiment was conducted from May to October, in 2016 and 2017, at the Jiu Zhuang Ecological Park in Hetao Irrigation District, Inner Mongolia Autonomous Region, China (40°41′ N, 107°18′ E, 1042 m a.s.l.). The location of the experimental site is provided in Figure 1. This area is characterized as having a semi-arid continental climate of middle temperate, with an average annual rainfall of 140 mm and annual pan-evaporation exceeding 2032–3179 mm. The mean annual and maximum wind speed are 2.5–3.4 m/s. The mean annual air temperature is 6–8 °C. There are 130 frost-free days and 3229.9 h of sunshine per year. The main physical properties of the soil layers (0–100 cm) are listed in Table 1. Meteorological data for the two growing seasons is provided in Figure 2. Rainfall in 2016 was 15% less than the average rainfall, and rainfall in 2017 was only 1/3 of the average rainfall. The average air temperature was 20.84 °C in 2016 and 21.62 °C in 2017.

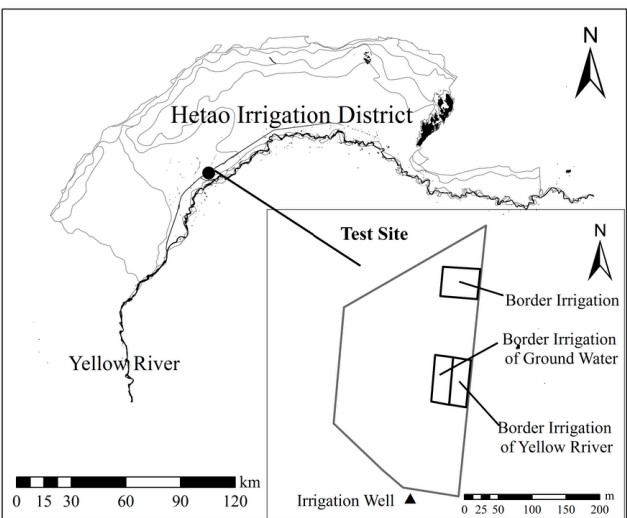

**Figure 1.** Test area position map.

**Table 1.** The physical properties of the tested soil.

| Depth (cm) | Bulk Density (g/cm$^3$) | Sand (2.0~0.02 mm) | Particle (<0.02~0.002 mm) | Clay Particles (<0.002 mm) | Classification of Soil Texture in the International System |
|---|---|---|---|---|---|
| | | Particle Mass Fraction at Different Levels (%) | | | |
| 0–90 cm | 1.39 | 24.31 | 62.09 | 13.6 | Silty loam |
| 90–120 cm | 1.54 | 87.78 | 11.16 | 1.06 | sand |

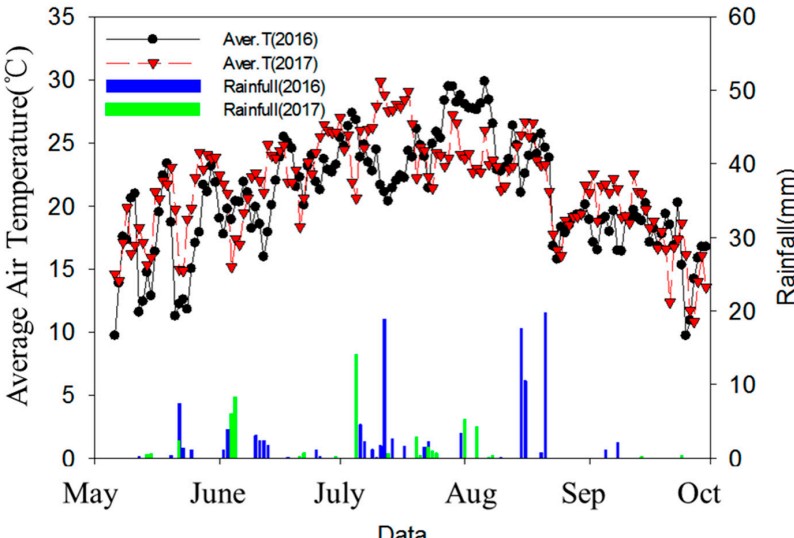

**Figure 2.** Average air temperature and rainfall distribution during the two growing seasons for maize.

Data source: these data come from experimental analysis. Before the start of the experiment in April 2016, soil samples were taken from depths of 0–10 cm, 10–20 cm, 20–40 cm, 40–60 cm, 60–90 cm, 90–120 cm, respectively, in the test site and each sample had three repeats. These soil samples were taken back to the laboratory, air-dried, filtered through a 2 mm sieve and tested by a laser particle size analyzer (HELOS/OASIS, New Patek Co., Ltd., Germany).

The soil of 0–90 cm from the test site is sandy loam, and 90–120 cm is sand.

### 2.2. Experimental Design and Field Management

This experiment set up three different irrigation methods and applied biochar for each method. Thus, the three combinations were border irrigation using Yellow River water with Biochar (YB), border irrigation using groundwater with Biochar (GB), and drip irrigation using groundwater with Biochar (DB). In addition, the three irrigation methods without biochar were used as a control and are referred to as border irrigation with Yellow River water (Y(CK)), border irrigation of groundwater (G(CK)), and drip irrigation of groundwater (D(CK)). In all treatments, the area of each plot was 180 m$^2$ (length 25 m × width 7.2 m). The six branes were treated with plastic mulch of 70 cm width and two lines of planting in each treatment. The distance between adjacent plants was 25 cm, and the row distance was 60 cm. The crop was XiMeng6 (a common maize variety used by local farmers, and has the characteristics of drought and lodging resistance, high yield), and the planting density was 66,000 strains/hm$^2$. The crop was planted on 7 May and harvested on 8 October in 2016 and 2017, with a growing period of 132 days.

The straw biochar used in the experiment came from the JinHeFu Agriculture Development Company, in Liao Ning Province. The straw biochar was applied to the surface and mixed with soil to 20 cm depth using a vertical rotary tiller in 2016. A total of 30 t/hm$^2$ straw biochar was applied at each treatment. The straw biochar was not used in 2017. The main properties of the straw biochar are as follows: particle size range 1.5~2.5 mm (>2 mm) is 60%; density 596 kg/m$^3$; pH 9.28; total C 72.21%; total N 1.56%; total P 0.72%; total K 1.64%.

The quantity of irrigation water of GB, G(CK), YB and Y(CK) used was consistent with that used by local farmers. Irrigation was conducted on 27 June, 16 July and 3 August in 2016 and on 17 June, 12 July and 31 July in 2017. A water meter was used to record the quantity of border irrigation used, and each irrigation quota was 150 mm. The lower limit of irrigation water for drip irrigation was −30 kPa, which was controlled by the tension meter to control the soil matrix potential 20 cm below the drop head. Irrigation was started when the tension meter reached −30 kPa and poured 22.5 mm of water. The date and amount of drip irrigation are provided in Figure 3. The drip irrigation tape

selected was the inner mounted drip irrigation belt produced by the Shanghai Huawei Company, and the pipe diameter was 16 mm. The drip-head flow rate was 1.38 L/h under the standard of 0.1 MPa. The distance between two drip-heads was 30 cm. The drip irrigation belt was laid in the middle of the two rows of corn along the corn direction, and each drip irrigation belt controlled two rows of corn. The temperature of groundwater is different from the Yellow River water. The change in temperature in the two years of the study is provided in Figure 4.

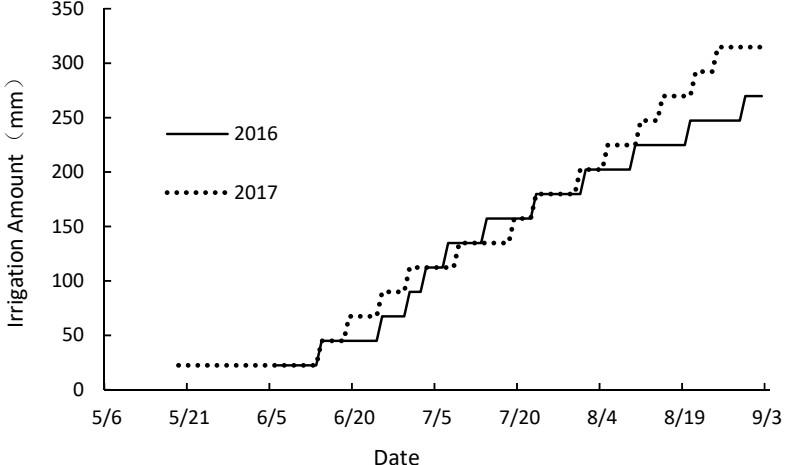

**Figure 3.** Accumulative drip irrigation water during 2016 and 2017.

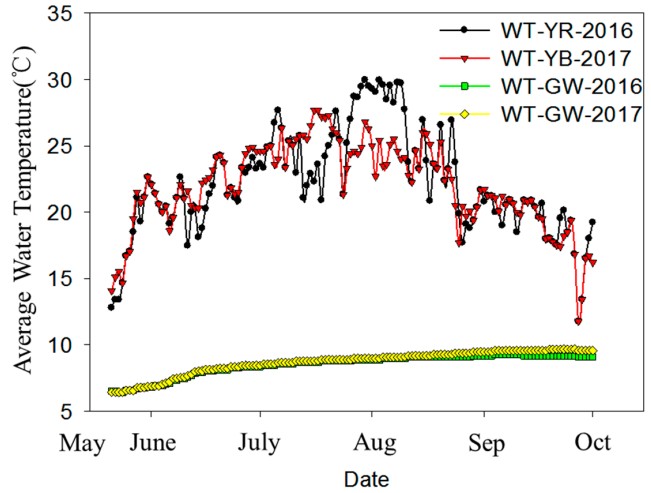

**Figure 4.** The temperature of irrigation water.

### 2.3. Data Acquisition

The soil temperature was automatically obtained using a multi-point soil temperature meter (YM-04/Yi-Meng Electron Ltd. of Handan/China, the manufacturer has calibrated the instrument.). Because the surface soil is exposed to the atmosphere and is more affected by solar radiation, its temperature changes more dramatically than at depth. Nonetheless, the deeper soil temperature is still affected slightly by the atmosphere and the temperature change on the surface. In addition, biochar is only mixed in the top 20 cm of soil. The soil temperature was measured at 5, 10, 20 and 40 cm soil depths, both inside and outside of the membrane (in Figure 5). In addition, the temperatures of the Yellow River and groundwater were monitored automatically by level loggers (HOBO-U20, USA). Meteorological data were collected by automatic weather stations installed in the field. The soil temperature was measured from 17 May to 7 October in 2016 and 2017. The location of the soil temperature probe is provided in Figure 5.

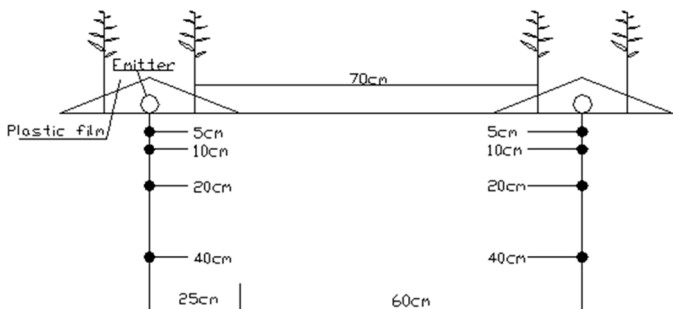

**Figure 5.** Mapping of soil temperature testing instruments.

Soil data were continuously monitored from 15 May to 15 October in 2016 and 2017. Data were obtained from 20 May to 8 October in 2016 and 2017 to avoid errors caused by the installation of instruments. The data was recorded automatically once every hour. In this study, soil temperature of temporal and spatial variation characteristics were analysed using hourly data. Other analytical data was calculated using an average value of 24 h in a day.

*2.4. Calculations and Data Analysis*

To analyse and compare the effects of different irrigation methods on soil temperature, the coefficient of different soil temperatures is defined using the mean of the variance of the average soil temperature sequence during the same treatment at different depths during the trial period. From a mathematical point of view, variance reflects the deviation degree and fluctuation degree between the numbers. If the mean variance composed by a serial number of average daily temperatures is larger, this indicates a greater fluctuation of soil temperature, which thereby indicates a greater difference in temperature.

$$\lambda_i = \frac{1}{n} \sum_{j=1}^{n} \left( Var\left(T_{jik}\right)\right)_j \tag{1}$$

where $\lambda_i$ represents the difference in the coefficient of soil temperature in the $i$th treatment; $T_{jik}$ represents the daily average temperature on the $k$th day in the $i$th treatment at $j$th cm soil layers; $i$ represents the soil layers at depths of 5, 10, 20 and 40 cm in DB, D(CK), YB, Y(CK), GB, and G(CK) treatments; $j$ represents the different treatments; and $k$ represents the day on which the soil temperature was measured.

## 3. Results

*3.1. Soil Temperature Difference under Various Treatments*

In this study, the soil temperature of different treatments varies with the season, as shown in Figures 6 and 7. With a change in air temperature, soil temperature changes in the form of a sine curve. In deeper soil, the peak value decreases. The soil temperature of both treatments was at its highest levels in late June and early July. The average soil temperature of border irrigation (19.0 °C) was higher than that of drip irrigation (18.65 °C) in the two growing seasons.

The variation coefficient of the soil temperature under different treatments, according to Equation (1), is shown in Table 2. The variance at a soil depth of 5 cm is the greatest under the same irrigation method. The variance decreases in deeper soil. This means that in deeper soil, the temperature fluctuated slightly. The variance of soil temperature under the film was larger than that outside the film under drip irrigation, and the variance under the film was smaller than that outside of the film under the border irrigation. The disparity in results between these two irrigation methods was due to their differing amounts of irrigation applied. It was also found that the fluctuation of soil temperature under border irrigation was larger than that under drip irrigation. This is mainly due to the long interval used in border irrigation and the effect on water temperature, particularly lowering the temperature of groundwater, when irrigation occurred. The maximum and minimum values of

λ were 11.19 and 6.47 for the treatment of G(CK) and drip irrigation of groundwater, respectively. The change in the λ value indicates that the fluctuation of soil temperature under G(CK) is large, with a maximum difference (λ = 3.76) in two years. With biochar application (D(CK), Y(CK), G(CK)), the λ decreased, which indicates that the fluctuation of soil temperature apparently decreased after the application of biochar in the groundwater border irrigation.

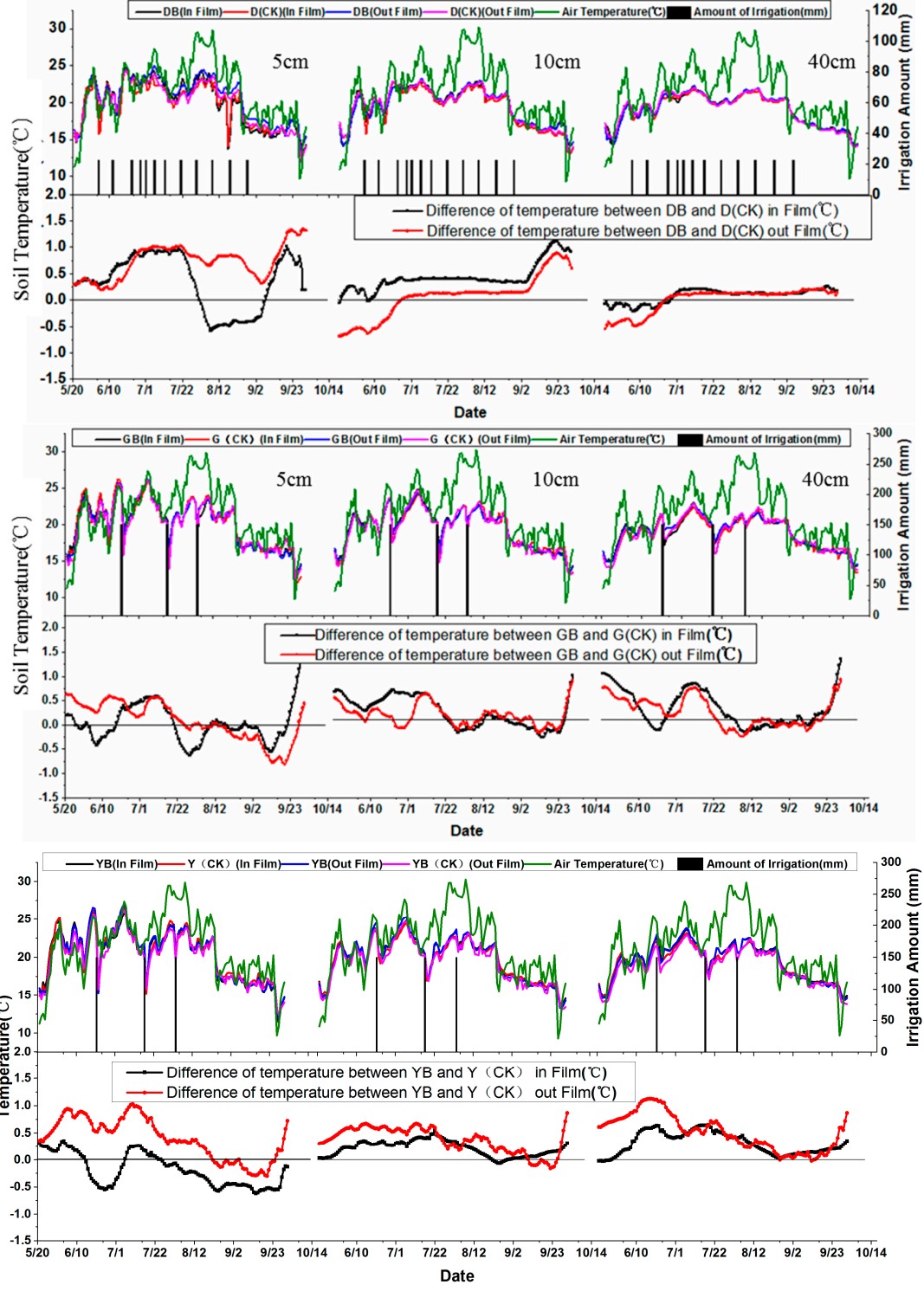

**Figure 6.** Seasonal dynamic changes of soil temperature in 2016. Note: 5 cm, 10 cm and 40 cm represent of the soil depth of 5 cm, 10 cm and 40 cm.

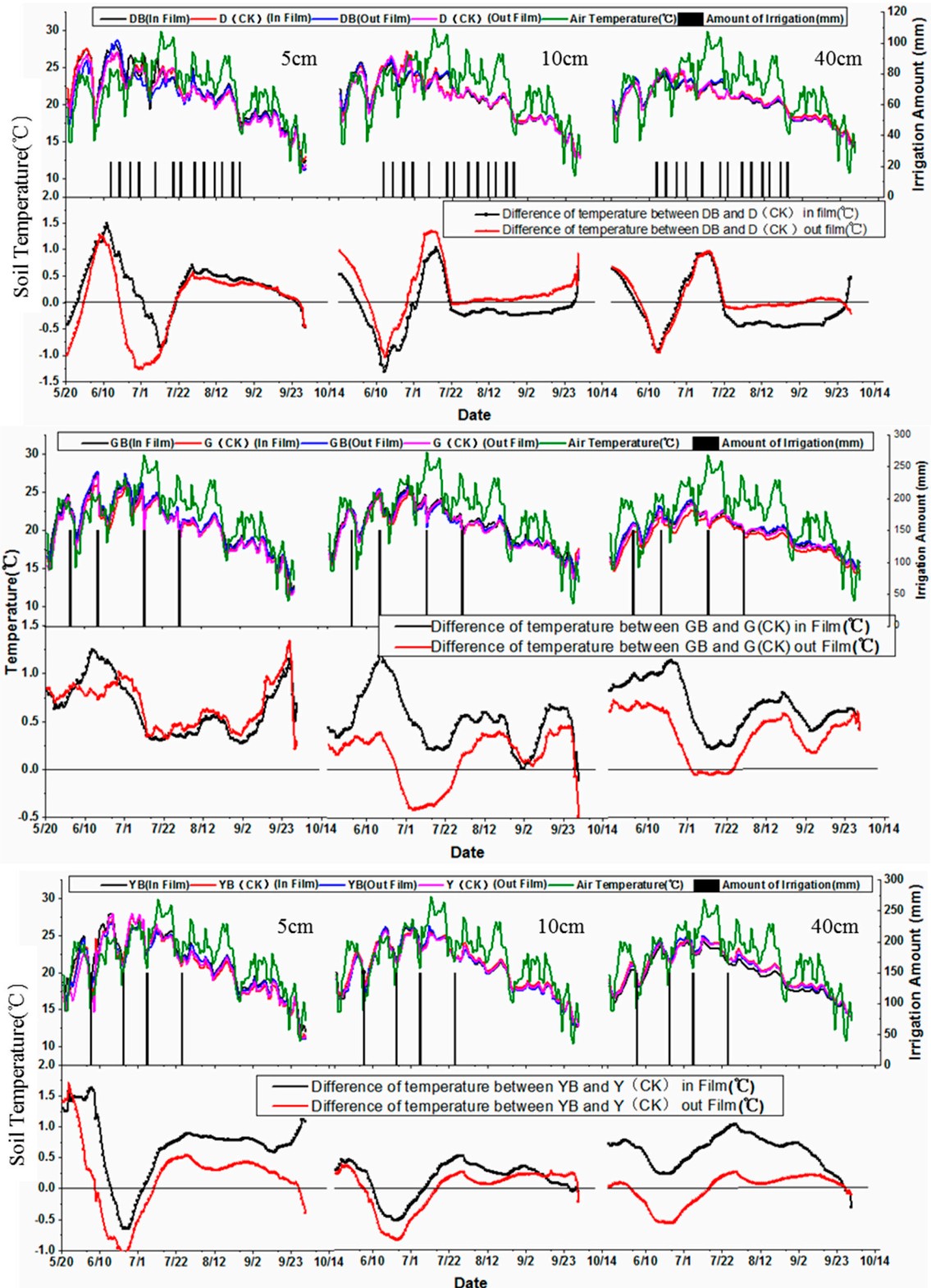

**Figure 7.** Seasonal dynamic changes of soil temperature in 2017. Note: 5 cm, 10 cm and 40 cm represent the soil depth of 5 cm, 10 cm and 40 cm.

**Table 2.** Temperature difference coefficient of different treatments.

| Treatments | Soil Temperature (°C) | Soil depth (cm) (2016) | | | | | | | | λ | |
|---|---|---|---|---|---|---|---|---|---|---|---|
| | | 5 | | 10 | | 20 | | 40 | | | |
| | | Intramembrane | Outer Membrane | Intramembrane | Outer Membrane | Intramembrane | Outer Membrane | Intramembrane | Outer Membrane | Intramembrane | Outer Membrane |
| DB | Max(T)/°C | 28.83 | 29.35 | 25.54 | 24.80 | 23.47 | 23.39 | 24.47 | 22.33 | 6.62 | 6.58 |
| | Min(T)/°C | 10.22 | 11.07 | 13.15 | 12.90 | 13.52 | 13.63 | 13.61 | 13.85 | | |
| | Temperature variance | 9.68 | 9.05 | 6.68 | 7.17 | 5.93 | 5.68 | 4.21 | 4.40 | | |
| D(CK) | Max(T)/°C | 27.36 | 27.53 | 24.55 | 25.03 | 23.02 | 23.26 | 21.88 | 22.10 | 6.68 | 6.77 |
| | Min(T)/°C | 10.17 | 10.37 | 11.81 | 11.69 | 13.03 | 13.24 | 13.95 | 13.83 | | |
| | Temperature variance | 8.97 | 8.79 | 7.25 | 7.55 | 6.07 | 6.04 | 4.45 | 4.69 | | |
| YB | Max(T)/°C | 31.16 | 30.38 | 26.54 | 27.56 | 25.01 | 26.74 | 24.20 | 25.96 | 7.94 | 8.98 |
| | Min(T)/°C | 7.68 | 9.48 | 11.91 | 11.50 | 13.56 | 12.96 | 14.37 | 14.12 | | |
| | Temperature variance | 11.16 | 11.88 | 8.33 | 9.74 | 6.71 | 8.13 | 5.56 | 6.18 | | |
| Y(CK) | Max(T)/°C | 30.94 | 32.46 | 26.63 | 30.17 | 24.69 | 26.47 | 23.31 | 24.68 | 7.37 | 8.93 |
| | Min(T)/°C | 9.50 | 9.19 | 12.06 | 10.66 | 13.56 | 12.88 | 14.25 | 14.44 | | |
| | Temperature variance | 10.64 | 12.49 | 8.11 | 9.39 | 6.05 | 8.31 | 4.68 | 5.54 | | |
| GB | Max(T)/°C | 29.35 | 29.17 | 26.86 | 26.40 | 25.29 | 24.70 | 23.16 | 22.97 | 7.21 | 7.46 |
| | Min(T)/°C | 10.38 | 9.78 | 12.12 | 11.77 | 13.40 | 13.27 | 14.04 | 14.25 | | |
| | Temperature variance | 9.91 | 10.71 | 8.13 | 8.39 | 6.42 | 6.29 | 4.38 | 4.44 | | |
| G(CK) | Max(T)/°C | 30.26 | 34.46 | 28.64 | 34.17 | 25.48 | 32.47 | 22.33 | 34.68 | 11.19 | 9.93 |
| | Min(T)/°C | 8.88 | 10.19 | 12.32 | 11.66 | 13.23 | 12.88 | 12.91 | 13.44 | | |
| | Temperature variance | 16.37 | 11.49 | 10.37 | 10.39 | 9.14 | 9.31 | 8.89 | 8.54 | | |

| Treatments | Soil Temperature (°C) | Soil depth (cm) (2017) | | | | | | | | λ | |
|---|---|---|---|---|---|---|---|---|---|---|---|
| | | 5 | | 10 | | 20 | | 40 | | | |
| | | Intramembrane | Outer Membrane | Intramembrane | Outer Membrane | Intramembrane | Outer Membrane | Intramembrane | Outer Membrane | Intramembrane | Outer Membrane |
| DB | Max(T)/°C | 28.95 | 30.68 | 26.99 | 28.3 | 25.99 | 27.88 | 24.3 | 24.81 | 8.87 | 8.15 |
| | Min(T)/°C | 10.04 | 9.38 | 10.53 | 10.19 | 10.24 | 10.82 | 11.79 | 12.41 | | |
| | Temperature variance | 12.35 | 10.45 | 10.01 | 9.83 | 7.99 | 8 | 5.14 | 4.33 | | |
| D(CK) | Max(T)/°C | 27.88 | 33.82 | 24.97 | 32.03 | 23.9 | 29.02 | 22.09 | 26.39 | 6.47 | 7.09 |
| | Min(T)/°C | 10.08 | 7.97 | 11.3 | 10.39 | 11.15 | 10.12 | 13.61 | 10.44 | | |
| | Temperature variance | 10.53 | 6.8 | 7.21 | 10.32 | 5.42 | 5.81 | 2.73 | 5.44 | | |
| YB | Max(T)/°C | 31.85 | 30.95 | 26.73 | 27 | 25.61 | 28.73 | 23.85 | 25.48 | 9.6 | 10.68 |
| | Min(T)/°C | 10.96 | 7.4 | 10.39 | 9.64 | 10.73 | 9.92 | 10.16 | 9.96 | | |
| | Temperature variance | 14.34 | 16.7 | 10.08 | 11.07 | 8.24 | 9.27 | 5.75 | 5.69 | | |
| Y(CK) | Max(T)/°C | 30.61 | 33.3 | 27.03 | 30.23 | 24.62 | 27.33 | 23.86 | 24.99 | 9.58 | 9.78 |
| | Min(T)/°C | 9.98 | 8.22 | 10.94 | 10.45 | 12.9 | 9.03 | 14.46 | 8.79 | | |
| | Temperature variance | 15.6 | 16.1 | 10.79 | 12.5 | 6.88 | 6 | 5.04 | 4.52 | | |
| GB | Max(T)/°C | 29.98 | 31.43 | 26.99 | 29.49 | 25.99 | 26.61 | 24.81 | 24.4 | 8.57 | 7.9 |
| | Min(T)/°C | 10.07 | 13.03 | 10.86 | 15.18 | 9.46 | 16.96 | 10.58 | 16.99 | | |
| | Temperature variance | 12.31 | 8.88 | 8.69 | 7.67 | 7.64 | 7.47 | 5.65 | 7.58 | | |
| G(CK) | Max(T)/°C | 29.88 | 33.3 | 28.29 | 30.23 | 25.89 | 27.33 | 22.77 | 24.99 | 10.68 | 8.73 |
| | Min(T)/°C | 8.43 | 7.22 | 11.92 | 10.45 | 13.47 | 9.03 | 14.2 | 8.79 | | |
| | Temperature variance | 16.15 | 13.94 | 12.4 | 10.21 | 8.14 | 6.67 | 6.03 | 4.12 | | |

However, the value of λ in drip irrigation and the Yellow River water border irrigation with biochar was higher than that in the treatments without biochar, which may be caused by the low temperature of groundwater and the evident thermal insulation of the biological carbon. In addition, the soil temperature under drip irrigation and the Yellow River water border irrigation was more affected by the air temperature and solar radiation. This may be because of the low water temperature of groundwater and the possibly significant role that biochar plays in heat preservation. Meanwhile, the soil temperature under drip irrigation and the Yellow River water border irrigation are more affected by the air temperature and solar radiation.

Compared with the analysis of temperature in the treatments with and without biochar, the soil temperature increased after biochar application. In the entire growth period, the temperature difference ranged from −0.7 °C to 1.1 °C under irrigation. The average temperature was 0.2 °C, 0.28 °C and 0.3 °C in drip irrigation, Yellow River water border irrigation, and groundwater border irrigation, respectively.

### 3.2. Temporal and Spatial Distribution of Soil Temperature under Various Treatments

Using the soil temperature on 25 June under different treatments as an example, the daily variation characteristics of soil temperature in the soil profile are analysed in Figure 8. The highest temperature using biochar and either drip irrigation, border irrigation using groundwater or Yellow River were 21.44 °C, 23.18 °C and 23.56 °C, respectively, which are higher than the respective temperatures of 20.43 °C, 22.62 °C and 23.20 °C under the treatments without biochar. The highest temperature under different treatments was found in the 0–10 cm soil layer. Under drip irrigation, the daily soil temperature reached the highest and lowest temperature at 12:00–14:00 and 8:00–10:00, respectively. The lowest temperature of the soil profile under DB was lower than that under D(CK) during the same period, with the same trend observed for the two years studied. Under the border irrigation using Yellow River water and groundwater, the daily soil temperature reached the highest temperature at 12:00–14:00. The maintain time for the high temperature in the above two treatments was longer than that under drip irrigation for 3 h and 2 h, respectively. In the vertical direction, in deeper soil, the density of the equivalent line decreased, but the density under drip irrigation was lower than that under border irrigation using Yellow River water and groundwater. Thus, under the condition of drip irrigation, the temperature of the deep soil maintained a relatively constant temperature for a long period of time.

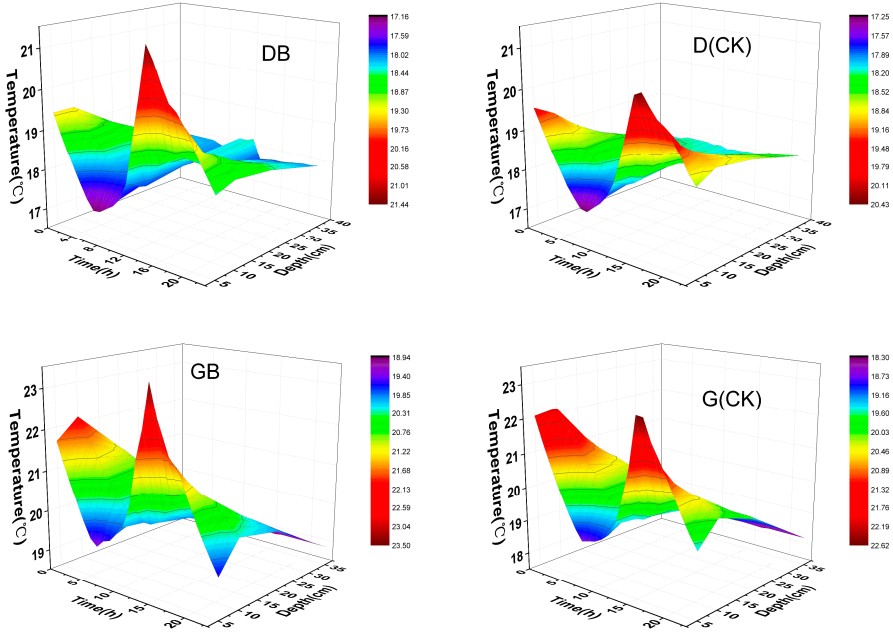

**Figure 8.** *Cont.*

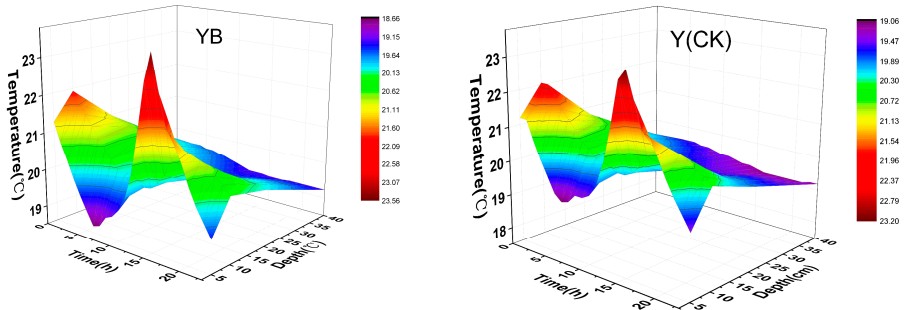

**Figure 8.** Temporal and spatial variation characteristics of soil temperature on different treatments.

Under the same irrigation method, the maximum temperature was reached 1 day earlier with biochar than without. It also cooled more sharply with biochar than under treatments without biochar when the atmospheric temperature decreased (Figure 9). This phenomenon was evident in the drip irrigation treatment, as the soil temperature in the biochar treatment increased 1 h earlier and reached the maximum 2 h earlier. Comparing the results of drip irrigation with and without biochar, the maximum temperature difference was 1.87 °C. The border irrigation using Yellow River water and groundwater showed a similar tendency, but the time at which the temperature fluctuated was different. The soil temperature under border irrigation using Yellow River water fluctuated at 13:30 and 17:00, while the soil temperature under border irrigation using groundwater fluctuated at 4:30 and 15:30. The amplitude of fluctuation was less than that of drip irrigation. The lag point of all treatments' wave-breaking point appears with deeper soil.

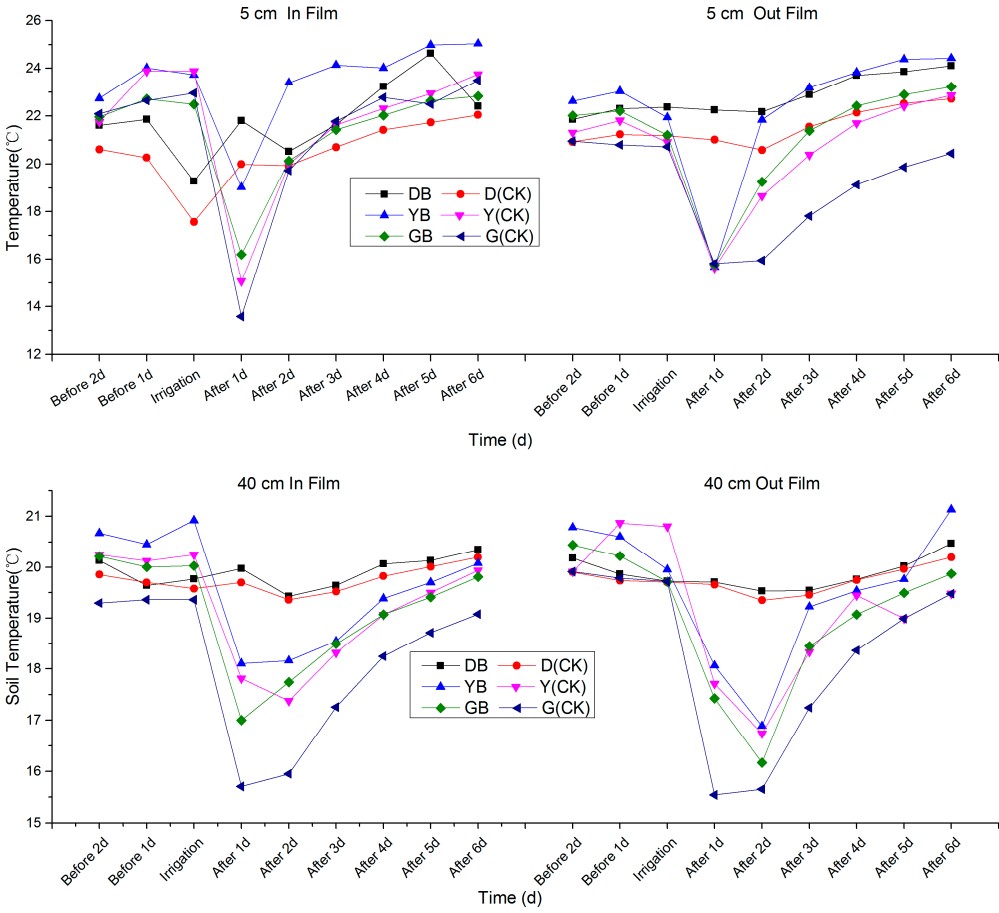

**Figure 9.** Effects of irrigation on soil temperature at different treatments and depths in 2016.

### 3.3. Change in Soil Temperature during Irrigation Period under Various Treatments

During the two years studied, the soil temperature of all treatments decreased after irrigation (Figures 9 and 10). In deeper soil, the soil temperature decreased slightly and took longer to recover to the original temperature before irrigation compared to surface soil. Under drip irrigation, only the soil temperature at 5–10 cm was affected by irrigation. The decrease in soil temperature was greatest 4 h after irrigation, with the highest value reduced by 3.05 °C. In deeper soil (5 cm, 10 cm, 20 cm, 40 cm), the maximum temperature of border irrigation using Yellow River water and groundwater decreased by 6.69 °C, 4.67 °C, 4.19 °C, 3.31 °C and 8.51 °C, 5.37 °C, 5.19 °C, 4.42 °C, respectively. The soil temperature gradually recovered to the original temperature before irrigation after 10 h with drip irrigation. With the deepening of soil depth, soil temperature is less affected. However, the soil temperature at 10 cm, 20 cm and 40 cm under border irrigation using Yellow River water and groundwater recovered to the original temperature before irrigation in 1 day, 2 days and 3 days, respectively. In the same soil layer, the change in soil temperature varied with the different treatments.

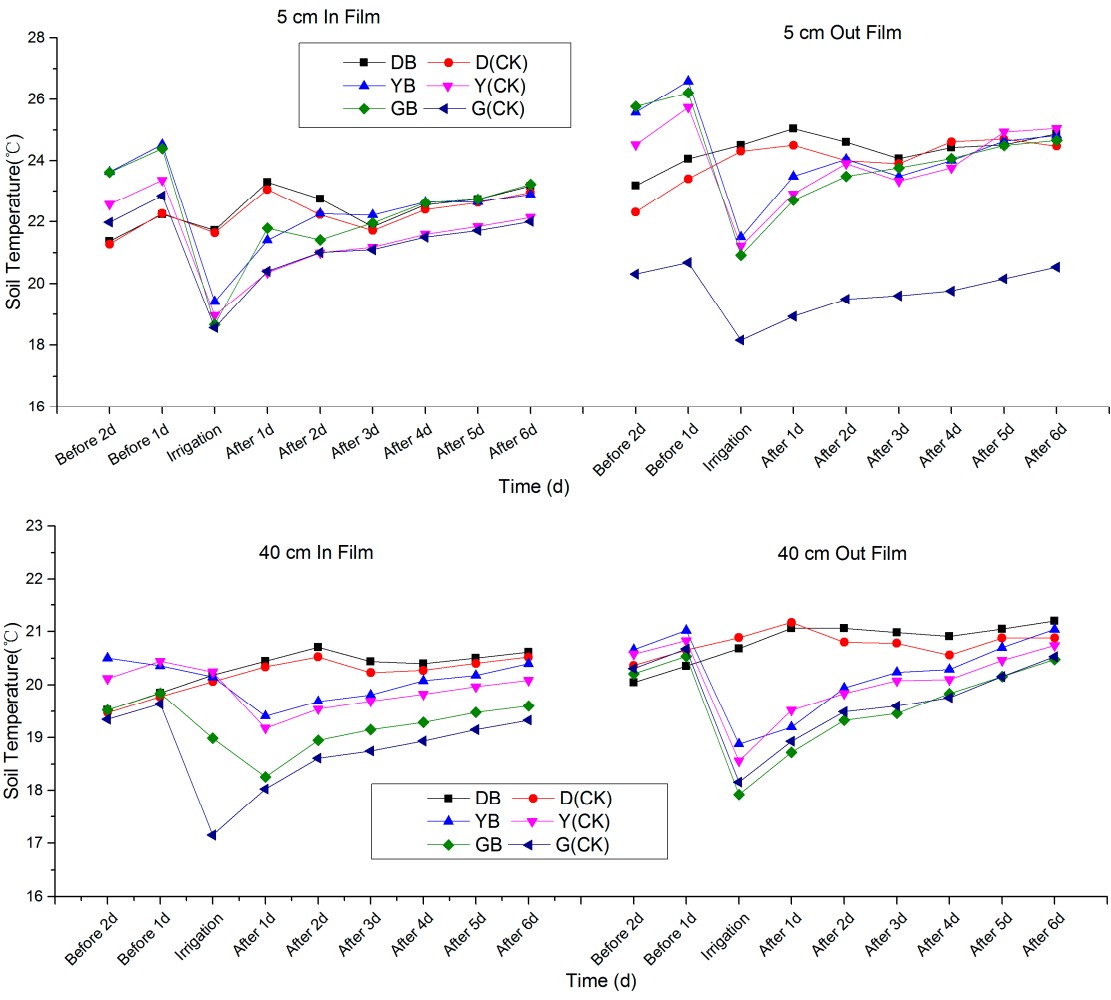

**Figure 10.** Effects of irrigation on soil temperature at different treatments and depths in 2017.

In 5 cm soil, the soil temperature of border irrigation (Y(CK), G(CK)) decreased, lagging by 12 h and 14 h, respectively, compared with that under drip irrigation. Soil temperature recovery lagged by 2 days, which was mainly due to the amount of water applied under border irrigation. At 10, 20, and 40 cm, the soil temperature under Y(CK) and G(CK) declined at the same time, but the soil temperature under Y(CK) recovered 1 day in advance compared with that under G(CK), which was due to the low water temperature of groundwater irrigation.

Under the same irrigation method, the soil temperature under treatments with biochar was higher than that without biochar. The soil temperature of treatments with biochar fluctuated slightly more than that without biochar after irrigation.

The soil temperature under DB and D(CK) reached the lowest temperature at the same time, but the largest temperature difference of DB was 0.72 °C lower than that of D(CK). At the 5 cm soil layer, the soil temperature of YB and Y(CK) reached the lowest temperature at the same time as that of G(CK), but the soil temperature of GB reached the lowest temperature 2 h before the equivalent in the 10 cm soil layer. The soil temperature of YB and GB reached the lowest temperature before that of Y(CK) and G(CK). At the depth of the 20 cm and 40 cm soil layer, although the soil temperature of YB decreased earlier than that of Y(CK), the soil temperature of YB and Y(CK) reached the lowest temperature at the same time. The soil temperature of YB and Y(CK) recovered 2 h earlier than that of GB and G(CK) due to the higher water temperature of Yellow river water than groundwater. However, the soil temperature of G(CK) decreased for a longer period than that of GB, and the recovery of the soil temperature under GB lagged.

*3.4. Characteristics of Soil-Accumulated Temperature during Crop Growing Period under Various Treatments*

Soil temperature is one of the indexes related to the soil heat condition, which varies with topography, soil moisture and weather. Soil temperature is positively related to the growth of maize within a certain temperature range. The accumulated soil temperature is the summation of the daily average temperature of a soil layer during a period of time. The irrigation method used and the application of biochar carbon have a major influence on the accumulated temperature. The characteristics of the accumulated temperature vary across soil depth profiles. The changes in the accumulated soil temperature in each soil layer during the growing period are shown in Figure 11. The accumulated soil temperature changed according to the same trend as seen in the seeding, jointing and filling stages. The maximum value of the accumulated soil temperature appeared at 5 cm depth and then decreased with deeper soil, reaching the minimum value in 40 cm soil. The accumulated soil temperature varied in the different growing stages, and the change rates of the accumulated temperature in each growing period were different. The largest decreasing rate was 19.64% in the seedling stage and the lowest was 0.2% in the maturity stage. This indicates that the accumulated soil temperature decreased mainly in the seedling stage of maize, and that almost no change occurred in the maturity stage due to the lack of irrigation. The accumulated soil temperature in the jointing stage was clearly greater than that in other growing stages, which was mainly due to the longest duration occurring in the jointing stage. The maximum values of the accumulated soil temperature under DB, D(CK), YB, Y(CK), GB and G(CK) during the entire growth period for the two years were 2883.87 °C, 2843.02 °C, 2784.39 °C, 2728.62 °C, 2732.81 °C and 2698.99 °C, respectively. It can be seen that the greatest increase in accumulated soil temperature occurred under DB, indicating that applying biochar with drip irrigation benefits the crop.

At the seedling stage, the accumulated temperature of each treatment declined in a straight line. The linear trend of DB and D(CK) treatment decreased between 0~10 cm, and the declining trend at 20–40 cm evidently became gentler. The temperature of YB and Y(CK) decreased linearly in 0~20 cm soil and declined gently at 20~40 cm, while the soil temperature of GB and G(CK) decreased linearly. During the jointing stage, in the 0~20 cm soil layer the soil temperature in all treatments decreased linearly, except for drip irrigation. A similar trend was observed in the jointing and heading stages. The accumulated soil temperature of each layer in the filling stage was virtually identical as well. In the maturity stage, the soil accumulated temperature increased linearly with the increase in soil depth. The soil temperature of treatments applying biochar declined slightly compared to that of treatments without biochar, offering further confirmation that biochar carbon has a buffering effect on the change in soil temperature.

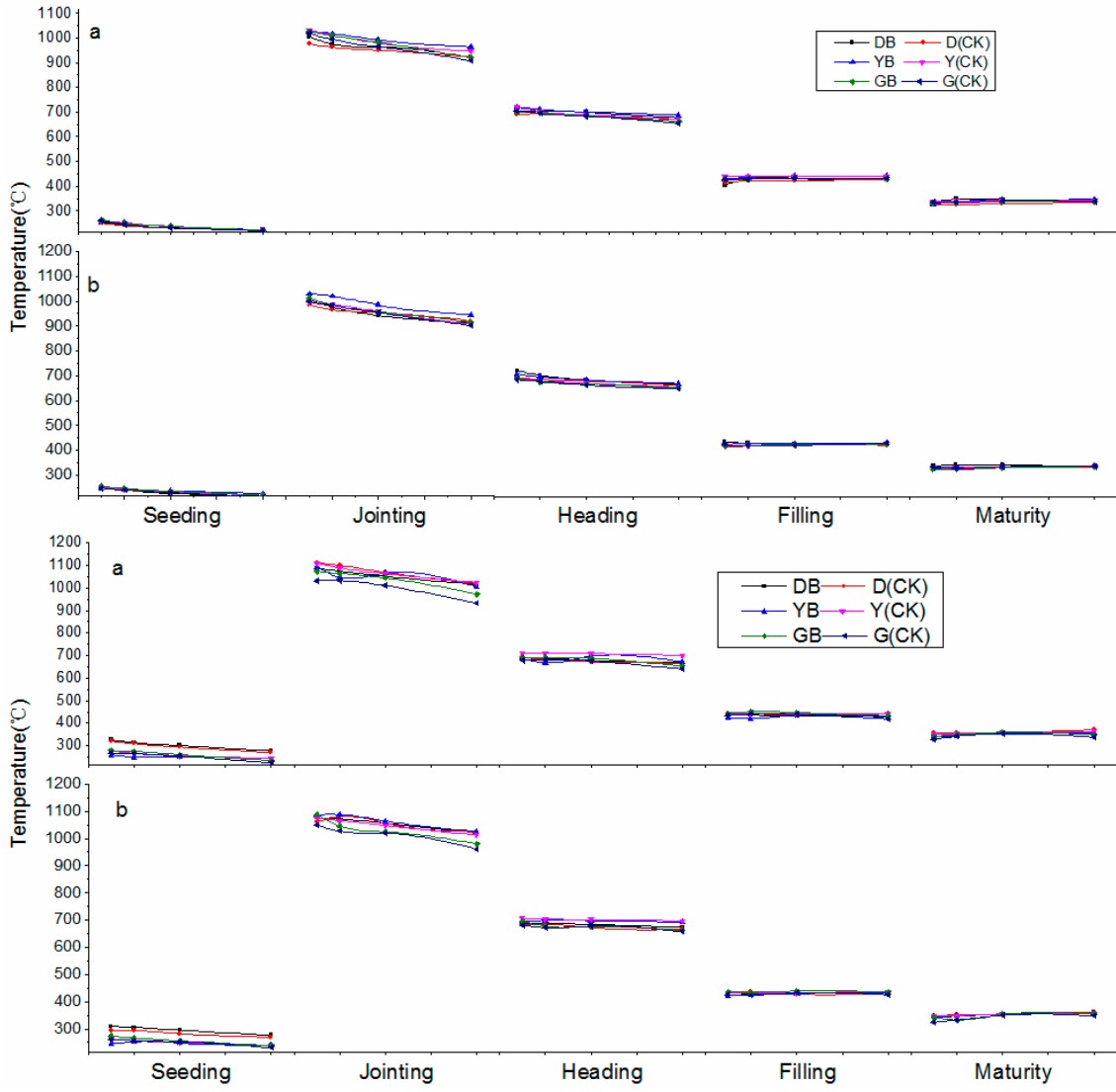

**Figure 11.** Characteristics of accumulated temperature change.

## 4. Discussion

### 4.1. Effects of Irrigation on Soil Temperature under Various Treatments

In farmland, soil temperature is not only affected by solar radiation and air temperature but also by irrigation water temperature, water quantity and frequency. In this study, which was carried out from early May to mid-June during the seeding stage of maize in the experimental site, the air temperature was low and the canopy density was close to zero. Thus, the soil temperature was primarily affected by solar radiation. The canopy density was greater than 90% from late June to late August, so the soil temperature (14.19 °C to 27.00 °C) in this stage was primarily affected by air temperature. From late August to October, as a result of the plant leaves withering, crop canopy density decreased, and the soil temperature was affected by air temperature and solar radiation. When the border irrigation was applied, the soil temperature was affected by the temperature of the irrigation water.

Long wave radiation is reflected back into the air near the ground, which is transferred to heat energy. This increases the air temperature around the crops. Short-wave radiation is reflected by the surface, which converts it into heat to increase the soil temperature. The other short waves were transmitted to the deep soil by heat conduction and raised the soil temperature. Film mulching decreases soil evaporation and maintains soil moisture to reduce the heat loss by evaporation [25,26].

After irrigation, soil humidity increases and the surface reflection rate decreases, which will increase the specific heat capacity of soil and significantly delay capacity [27]. In the Hetao irrigation district, the soil temperature is mainly influenced by irrigation during the crop growing period. Due to the different supply conditions of soil heat caused by soil humidity, the effect of irrigation methods on soil temperature varies. Irrigation methods not only determine the distribution of soil moisture but also significantly affect the soil micro-climate such as humidity and temperature [28]. The border irrigation using Yellow River water and groundwater is a type of ground irrigation, which requires reduced application times and generates greater quantities of water than other methods. Additional irrigation changes the soil water distribution, which results in a different soil heat distribution [29]. In this study, under drip irrigation, the surface soil temperature first decreased after irrigation and then quickly recovered to the original temperature. This was because frequent irrigation keeps the soil temperature within a relatively stable range. Therefore, drip irrigation is of great significance to efforts related to managing soil temperature [30]. There were long intervals between every two irrigation applications in the border irrigation using the Yellow River water and groundwater. When no irrigation was applied, the soil temperature was mainly influenced by solar radiation and air temperature. From early May to early June, the crop canopy density was nearly zero, and the soil temperature was affected by solar radiation. In the daytime, soil is affected by radiation. According to the Second Law of Thermodynamics [31], the soil is in an endothermic state during the day, while it is in an exothermic state when the temperature is reduced at night. From mid-June to late September, crop canopy density increased and soil temperature was mainly affected by air temperature. When irrigation was applied, the soil temperature was mainly determined by the temperature of the irrigation water.

Although the soil humidity increases and the surface reflectance decreases during irrigation, large amounts of water entering the soil over a short time period will increase the soil's heat capacity. More heat is then required to increase the temperature. The heat energy converted by solar radiation cannot compensate for the low temperature driven by the irrigation water temperature over a short time period. Guo-Hua et al. [15] reported that ground border irrigation, sprinkler irrigation and drip irrigation had changed the profile distribution of the soil temperature. The high-frequency irrigation of the sprinkler irrigation and drip irrigation made the surface temperature of the farmland lower than that of the ground irrigation, so the temperature difference of the relevant section was small. Sun et al. [16] suggested that the surface temperature of drip irrigation was lower than that of groundwater irrigation and border irrigation in the Yellow River. The present study's findings are consistent with these related results.

Applying biochar to soil was beneficial for increasing soil temperature, reducing surface emissivity, and increasing soil humidity [32,33]. Li et al. [34] reported that applying biochar to soil increases soil porosity and the soil-water content. The results in this study demonstrated that applying biochar could effectively stabilize the fluctuation range of soil temperature. Soil temperature was mainly affected by solar radiation at the seedling stage and the later stage of the crop. The application of biochar was able to reduce the surface reflectance, and thus more heat was absorbed. Soil temperature decreased as air temperature increased, and applying biochar increased soil porosity, causing more heat to be lost to the atmosphere This indicates that applying biochar can reduce the heat entering the soil as well as the heat lost, which is consistent with the results of Liu et al. [34]. After the application of biological carbon, the soil temperature was more sensitive to external factors (air temperature and water temperature) because the biochar's black colour makes it more sensitive to solar radiation, which increases the temperature more rapidly. Furthermore, because applying biochar increases the water retention of the soil [34], the specific heat capacity of water was greater than that of the soil. Therefore, the soil temperature was lower with biochar treatment than without under higher temperatures. In the stage without irrigation, the 24-h change in soil temperature also exhibited this tendency, providing additional evidence that the application of biochar to soil hinders temperature change in two ways. This finding was not apparent in deeper soil layers. The temperature of deep soil was mainly influenced by the heat transferred from the surface soil, and the biochar helped preserve

heat at depth. Thus, there was no apparent change in the soil temperature in the deeper soil layers. After applying biochar under drip irrigation, a greater temperature difference was observed compared to that without biochar. In the daytime, the soil temperature with biochar was higher than that without biochar due to the effect of solar radiation. At night, the temperature was slightly lower than that without biochar applied. This temperature difference also reduced the ineffective respiration of crops at night, increasing the accumulation of organic matter in the crops. However, because of the long time interval between two irrigation periods, this tendency was not apparent in the YB and GB. This trend was not obvious under the condition of border irrigation using groundwater and Yellow River water.

Therefore, the application of biochar under drip irrigation can improve the soil temperature in the early stage of crop growth, reduce the soil temperature in the mid-stage of crop growth, and create a beneficial hydrothermal environment for the crops. Larger temperature differences between day and night were beneficial for the accumulation of organic matter in crops. The application of biochar under the condition of border irrigation using Yellow River and groundwater improved the temperature difference in soil, preserved heat at lower temperatures, and limited the temperature under high temperatures. At present, most of the farms in the Hetao irrigation area still use border irrigation, groundwater or Yellow River water. The application of biochar will help improve the water and heat environment of the farmland.

*4.2. Biochar and Irrigation Methods' Effect on Soil Accumulated Temperature*

The change in soil accumulated temperature is the most important factor in the process of plant growth. It is necessary to grasp the change in the law of soil accumulated temperature to understand the law of water demand in order to create suitable growth conditions for corn. Our results show that the soil accumulated temperature of border irrigation using groundwater and Yellow River water was lower than that of drip irrigation. Under the same irrigation method, the accumulated temperature of soil where biochar was applied was higher than that without biochar. The accumulated soil temperature of D(CK), G(CK), and Y(CK) increased by 15.32%, 13.20% and 10.28%, respectively, especially in the seeding stage, when biochar was applied. In the seeding, jointing and leading stages, the accumulated soil temperature was positively linear with soil depth and demonstrated an opposite trend in the mature stage. Therefore, the application of biochar under suitable irrigation methods is beneficial to the growth of maize, playing an important role in realizing water savings and in complementing synergistic irrigation methods.

## 5. Conclusions

It was found that applying biochar to the soil can improve the soil temperature of the plough layer. Coupling drip irrigation with biochar, soil temperature increased by 1.20~3.87%. Coupling border irrigation using Yellow River water and groundwater with biochar, soil temperature increased by 0.80~2.40% and 1.01~5.15%, respectively.

Biochar is a medium for reducing the heat exchange between the soil and the atmosphere, interfering with the bi-directional heat movement. This was especially apparent at the 0–10 cm soil layer in the border irrigation treatments using Yellow River water and groundwater. Biochar effectively stabilized the fluctuation of soil temperature and improved the soil accumulated temperature. These changes were mainly concentrated above the 10 cm soil depth. After applying biochar to the soil, soil temperature became more sensitive to external temperature changes, such as air temperature and water temperature.

Therefore, in the Hetao irrigation area, applying a proper amount of biochar to farmland soil can improve its water and heat environment and improve the synchronization of water and heat achieved through traditional border irrigation, especially using the drip irrigation method. Therefore, using biochar under drip irrigation can improve the performance of agricultural lands.

**Author Contributions:** Conceptualization, Y.D. and Z.Q.; methodology, Y.D.; software, X.G.; validation, Y.D., X.G. and Z.Q.; formal analysis, C.L.; investigation, Y.J.; resources, M.H.; data curation, Y.D.; writing—original draft preparation, Y.D. and X.G.; writing—review and editing, Y.D. and X.G.; visualization, Y.D.; supervision, Z.Q.; project administration, Z.Q.; funding acquisition, Z.Q. and X.G.

**Funding:** This work was jointly supported by the National Key Research and Development Plan (2016YFC0501301) and the National Natural Science Foundation of China (51779117, 51809142).

**Acknowledgments:** The authors thank Zhe Li and Shaodong Yang for help in data processing and paper writing.

**Conflicts of Interest:** The authors declare no conflict of interest.

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
