# Peer review of "Effects of Biochar Application and Irrigation Methods on Soil Temperature in Farmland"

_water, doi:10.3390/w11030499_

Round 1

Reviewer 1 Report

The authors present an experiment dealing with the changes in soil temperature after biochar application as modified by irrigation. The paper is well written (clear although not concise) and the methods suitable for the study goals. I have only minor suggestions to improve its quality. 

- First, the introduction is a bit confusing, since many concepts start, then left and again discessed later. I suggest to order this part.

- Some information lacks in the method section (see the annotated file in attachment).

- Some figures are difficult to be read. Their quality can be improved.

Author Response

Dear Editors and Reviewers:

Thank you very much for your attention and the objective evaluation. We are very grateful to your comments for the manuscript (Manuscript ID: water-454759), entitled" Effects of biochar application and  irrigation methods on soil temperature in farmland". According with your advice, we amended the relevant part in manuscript. Some of your questions were answered below.

Response to Reviewer 1 Comments

Point 1: The introduction is a bit confusing, since many concepts start, then left and again discessed later. I suggest to order this part.

Response 1: We have re-written this part according to the Reviewer’s comments. In order to make it easier for readers, we have adjusted the order of the content of the introduction, and revised and deleted some of the contents. The introduction is divided into five parts. The first part mainly talks about the importance of soil temperature. The second part mainly talks about some factors affecting soil temperature, and mainly explains from the aspects of solar radiation, soil moisture distribution and so on. The third part mainly talks about the research progress of the influence of irrigation methods on soil temperature. The fourth part mainly talks about the research progress of the influence of biochar on soil temperature at present. The fifth part mainly summarizes the shortcomings of the current research and points out the purpose of the research.

Details have been marked in red color in the manuscript.

Point 2:  Some information lacks in the method section (see the annotated file in attachment).

Response 2: We are very sorry for our negligence of some information lacks in the method section, which causes confusion to the editors and reviewers. We will add a description of the method in this study, and the detailed revisions are in the line 173 to line 174.

Point 3: Some figures are difficult to be read. Their quality can be improved.

Response 3: We have made correction according to editors’ and review's comments about figures. We modified Fig. 4, Fig. 6, Fig. 7, and the detailed revisions are in the line 195 to line 197; line 266 to 272 and line 273 to 281, and notes are added in Figures 6 and 7. the detailed revisions are in the line 273 and 282. We have added Fig. 5, which was omitted before, and the detailed revisions are in the line 214 to line 216. We changed the original fig. 9 to the present Fig. 9 and Fig. 10. We add a data source notes below Table 1, which in the line 148 to line 153; We deleted the Table 2, adding relevant content to the article, which in the line 178 to line 179, and changed the original Table 3 to the present Table 2, and revised the formats of Table 1 and Table 2.

Special thanks to you for your good comments.

We tried our best to improve the manuscript and made some changes in the manuscript. These changes will not influence the content and framework of the paper. And here we did not list the changes but marked in red in revised paper.

We appreciate for Editors/Reviewers’ warm work earnestly, and hope that the correction will meet with approval.

Once again, thank you very much for your comments and suggestions.

Reviewer 2 Report

the paper is interesting, sound and, in general, clearly structured and written. It also fits within the scope of the journal. Therefore, it might deserve publication already as it is.

Author Response

Thanks very much for your kind work and consideration of our paper. On behalf of my co-authors, we would like to express our great appreciation to editor and reviewers.

Thank you and best regards.

Reviewer 3 Report

This manuscript is well written.

Author Response

(The authors gave the same response as above.)
